# Meiotic Cell Cycle Progression in Mouse Oocytes: Role of Cyclins

**DOI:** 10.3390/ijms241713659

**Published:** 2023-09-04

**Authors:** Hye Min Kim, Min Kook Kang, Se Yoon Seong, Jun Hyeon Jo, Min Ju Kim, Eun Kyeong Shin, Chang Geun Lee, Seung Jin Han

**Affiliations:** 1Department of Biological Science, Inje University, Gimhae 50834, Republic of Korea; als42502@dirams.re.kr (H.M.K.); shin0032@dirams.re.kr (E.K.S.); 2Department of Research Center, Dongnam Institute of Radiological and Medical Sciences, Busan 46033, Republic of Korea; minkooking@dirams.re.kr (M.K.K.); cglee@dirams.re.kr (C.G.L.); 3Institute for Digital Antiaging Healthcare, Inje University, Gimhae 50834, Republic of Korea; sunis@oasis.inje.ac.kr (S.Y.S.); jjunhyeon98@oasis.inje.ac.kr (J.H.J.); kmj5395@oasis.inje.ac.kr (M.J.K.); 4Department of Medical Biotechnology, Inje University, Gimhae 50834, Republic of Korea; 5Institute of Basic Science, Inje University, Gimhae 50834, Republic of Korea

**Keywords:** oocyte maturation, cyclin, maturation promoting factor, translational regulation, protein modification

## Abstract

All eukaryotic cells, including oocytes, utilize an engine called cyclin-dependent kinase (Cdk) to drive the cell cycle. Cdks are activated by a co-factor called cyclin, which regulates their activity. The key Cdk–cyclin complex that regulates the oocyte cell cycle is known as Cdk1–cyclin B1. Recent studies have elucidated the roles of other cyclins, such as B2, B3, A2, and O, in oocyte cell cycle regulation. This review aims to discuss the recently discovered roles of various cyclins in mouse oocyte cell cycle regulation in accordance with the sequential progression of the cell cycle. In addition, this review addresses the translation and degradation of cyclins to modulate the activity of Cdks. Overall, the literature indicates that each cyclin performs unique and redundant functions at various stages of the cell cycle, while their expression and degradation are tightly regulated. Taken together, this review provides new insights into the regulatory role and function of cyclins in oocyte cell cycle progression.

## 1. Introduction

The statement “Omnis cellula-e-cellula” coined by Rudolf Virchow means “all cells come from cells”, and this concept forms the basis of the cell cycle. Based on this concept, the phrase “Omne vivum ex ovo” coined by William Harvey signifies the importance of the oocyte cell as the origin of life. In elucidating the key regulators of the cell cycle, including that of oocytes, cell division cycle (Cdc) genes involved in cell cycle progression were discovered in yeast during the 1960s. Subsequently, treating frog (*Rana pipiens*) oocytes with progesterone was found to induce cell cycle progression (oocyte maturation), and the injection of the cytoplasm of mature oocytes into immature oocytes promoted cell cycle progression without progesterone stimulation. The substance in the cytoplasm of the matured oocytes that promoted the cell cycle progression was named maturation-promoting factor (MPF) [1]. In 1983, Hunt’s group identified a protein stored as maternal mRNA in sea urchin (*Arbacia punctulate*) eggs that undergoes synthesis and degradation in a cell-cycle-dependent manner and named it cyclin [2]. Studies have revealed that MPF is a complex of cyclin and cyclin-dependent kinase 1 (Cdk1), also known as p34/cdc2, activated in a cyclin-dependent manner [3,4]. Subsequent experiments further identified several types of Cdks and cyclins that regulate various cell cycle points.

The cell cycle in multicellular eukaryotes is primarily regulated by Cdk1 and Cdk2, and animal cells have additional Cdks, such as Cdk4 and Cdk6, for controlling cell cycle entry in response to extracellular growth factors [5,6]. The human genome contains at least 20 Cdk genes. These Cdks function as kinases by generating specific Cdk–cyclin complexes at particular stages of the cell cycle. These complexes phosphorylate specific substrates to control cell cycle progression [7] or are involved in the regulation of transcription and other cellular processes [8].

Cyclins, which possess a well-conserved cyclin box domain (CBD), do not have enzymatic activity themselves, but provide a substrate-binding site for Cdks and direct Cdks to specific cellular locations. Various cyclins have been discovered, including 9–15 from fungi, 14 from flies and echinoderms, and more than 30 from human cells [6,9]. During cell cycle regulation, D-type cyclins regulate the transition from the G0 to G1 phase by binding to Cdk4 or Cdk6 [10,11], while E-type cyclins form complexes with Cdk2 and function during the late G1 and early S phases. Cyclin A2 binds to Cdk2, promoting both G1/S and S/G2 transitions and DNA replication, while regulating the initiation of mitosis by binding to Cdk1 during the G2/M transition [12]. The Cdk1–cyclin B complex, known as MPF, controls the progression of the M phase [13,14].

The meiotic process, a specialized cell division process (reduction division) where the chromosomes undergo two rounds of division after a single round of DNA replication to generate gametes with half the chromosome number, is also regulated by Cdk and cyclin complexes, similar to that of other somatic cells (Figure 1). Among gametes, oocytes are relatively large, have a relatively long cell cycle length, and show distinct cell cycle arrest and progression stages. In many organisms, oocytes are mostly in a state of cell cycle arrest and can only exit this arrest state by specific hormonal signals [15,16]. This fact suggests that oocytes require distinct and specific mechanisms, different from those of somatic cells, for the regulation of Cdk activity.

Mice lacking Cdk3, Cdk4, or Cdk6 are viable and undergo a normal oocyte maturation, indicating that these Cdks are not essential for oocyte maturation [17,18,19]. Cdk1 and Cdk2 expression has been observed in mouse oocytes, and oocyte-specific deletion of Cdk1 and Cdk2 showed that only the loss of Cdk1 prevents the resumption of meiosis in oocytes, leading to female infertility [20,21]. Therefore, Cdk1 is considered a key Cdk that regulates the resumption of meiosis in mouse oocytes [20,21,22].

Until recently, cyclin B1 was considered to be the primary cyclin component of the MPF complex, which regulates the meiotic cell cycle in oocytes. However, recent studies have reported the expression of various cyclins, including cyclins B2, B3, A1, A2, E, and O, in oocytes of various species, such as *Xenopus*, *Danio*, mice, and humans [23,24,25]. The specific roles of these cyclins in oocyte maturation have not been extensively detailed, and they were previously thought to have minor functions compared to cyclin B1 [26,27,28,29]. For example, cyclin B2 was presumed to be redundant with cyclin B1. However, recent biochemical and cell biological approaches and studies based on knockout mice have revealed that these cyclins play distinct and unique roles in oocyte maturation (Figure 1). For instance, recent research on the role of cyclin B3 in oocytes has led to increased interest in the roles of other cyclins besides cyclin B1 [27,28,29,30,31,32,33,34,35,36,37,38,39,40,41]. This review focuses on the roles of various cyclins in the regulation of the mouse oocyte cell cycle, with a particular emphasis on recent discoveries; discusses the roles of different cyclins in oocyte cell cycle progression; and examines the regulation of cyclin expression by translation and degradation in accordance with the sequential progression of the oocyte cell cycle.

### 1.1. Overview of Cyclins

More than 30 cyclins have been discovered in mice. Among them, the functions of cyclins A2, B1, B2, B3, and O in mouse oocytes have been studied (Figure 2) [27,28,29,30,31,32,33,34,35,36,37,38,39,40]. The fundamental features of these cyclins are discussed below.

#### 1.1.1. Cyclin B1

Synthesis of cyclin B1 begins during the S phase, and it exhibits activity during the G2/M transition after forming a complex with Cdk1. It is rapidly degraded during mitosis. In somatic cells, cyclin B1 is predominantly present in the cytosol but translocates into the nucleus when mitosis begins. It contributes to nuclear envelope breakdown, chromosome condensation, and mitotic spindle formation [42,43]. Deletion of *Ccnb1* (the gene encoding cyclin B1) results in embryonic lethality in the uterus, indicating its importance in mitotic progression. Furthermore, tissue-specific knockout studies have revealed its significance in the generation of oocytes and sperms [27,29,44]. Cyclin B1 has been traditionally considered to regulate all aspects of oocyte maturation. However, recent findings from literature searches demonstrate that cyclin B1 functions only at specific stages of oocyte maturation.

#### 1.1.2. Cyclin B2

Cyclin B2 is a subtype of cyclin B that is conserved in numerous vertebrate species [45]. In human tissue culture cells, cyclin B2 expression decreases during the G1 phase and increases during the S and G2 phases, reaching its peak during mitosis [46]. In *Xenopus laevis*, cyclin B2 is essential for cell cycle control during the G2/M transition and is steadily accumulated during the G2 phase before being rapidly degraded during mitosis [47,48]. Cyclin B2 in human culture cells is predominantly distributed in the Golgi apparatus and is evenly dispersed throughout the cell, unlike cyclin B1, which undergoes nuclear redistribution during prophase [43]. Cyclin B2, like cyclin B1, forms a complex with Cdk1 to activate it [43]. Knockout mice lacking *Ccnb2* (the gene encoding cyclin B2) are viable and develop normally [27,29].

#### 1.1.3. Cyclin B3

Cyclin B3, initially identified in chickens, is approximately 33% similar to cyclin B2 and approximately 30% similar to A-type cyclins [23,49]. In humans, cyclin B3 mRNA is expressed in all cell cycle stages, including the G0 phase, and is localized within the nucleus similar to A-type cyclins [49,50,51]. In *Drosophila* and *C. elegans*, cyclin B3 is found in mitotic and maternal germ cells. It regulates the S phase and promotes the metaphase-to-anaphase transition by alleviating the spindle assembly checkpoint (SAC) during mitosis [52,53,54,55,56]. In *Drosophila*, cyclin B3 is dispensable for mitosis but is necessary for female fertility [57]. In humans, cyclin B3 is most abundantly expressed in the testis and is found in all tissues [51]. Mice carrying a genetic knockout of *Ccnb3* (the gene encoding cyclin B3) show viability, indicating that cyclin B3 is dispensable for mitosis. For meiosis, males exhibit fertility but female mice are sterile [31,32,58,59].

#### 1.1.4. Cyclin A

Two forms of cyclin A have been identified in *Xenopus* [60], mice [61], and humans [62]. Cyclin A1 is considered to be the embryonic form and its expression in the germ cell lineage and in leukemia and brain cells is highly restricted. It is particularly highly expressed in the testes of mice and humans [61,62,63]. Moreover, it forms complexes with both Cdk2 and Cdk1 [64], and is involved in DNA double-strand break repair and ciliogenesis through its interaction with Cdk2 [65,66]. Male *Ccna1* (the gene encoding cyclin A1) knockout mice exhibit a block in spermatogenesis before the first meiotic division, resulting in sterility, while females are unaffected [34].

Cyclin A2 is the somatic form of cyclin A and is expressed in various tissues after embryonic development. Its expression begins during the S phase of the cell cycle and accumulates until prometaphase [61,67]. Cyclin A2 is important for DNA replication and mitosis, activating Cdk2 during G1/S transition and Cdk1 during mitosis [68,69]. It regulates entry into cell division and ensures faithful chromosome segregation by controlling kinetochore microtubules [70]. Cyclin A2 is rapidly degraded, independent of the SAC, from prometaphase onwards, prior to the degradation of cyclin B by the anaphase-promoting complex/cyclosome (APC/C) complex [68,71,72,73,74]. Deletion of *Ccna2* (the gene encoding cyclin A2) leads to embryonic lethality [75].

#### 1.1.5. Cyclin O

Cyclin O (also known as UNG2 or UDG2) is a cyclin-like DNA glycosylase involved in the removal of uracil misinserted or cytosine deaminated in DNA [76]. In human B cells, cyclin O binds to replicating chromatin in G1/S phase, specifically interacting with replication protein A, and is degraded in the G2 phase [77]. In mouse lymphoid cells, cyclin O associates with Cdk2 and activates it, leading to DNA damage-induced apoptosis [78]. In mice with genetic deletion of *Ccno* (the gene encoding cyclin O) multiciliated cells exhibit abnormalities in centriole amplification, resulting in a reduction in the number of multiple motile cilia [40].

## 2. Germinal Vesicle (GV) Arrest

Numerous mammalian females, including those of humans and mice, produce and store oocytes in the ovary during prenatal embryonic development. These oocytes undergo a meiotic process before birth and remain in prophase I (diplotene) of the cell cycle., The oocytes in small follicles (less than ~75 um in diameter) at this stage are meiotically incompetent and the cell cycle arrest is primarily due to insufficient levels of cell cycle promoting proteins, including Cdk1 [79]. Therefore, the meiotic arrest in this stage is not achieved through cAMP signaling or inhibition of Cdk1. As the follicles grow, typically reaching the primary follicle stage, the oocytes become meiotically competent, a stage that is characterized by large nuclei and is referred to as the germinal vesicle (GV) stage (Figure 1). The GV oocytes in the ovaries need to maintain this arrested state for an extended period of time, ranging from several months in mice to several decades in humans, before undergoing maturation and fertilization [80]. To ensure the completion of meiosis at the appropriate time, precise cell cycle regulation is required in the oocyte.

The prophase I arrest of the oocyte cell cycle is primarily regulated by high levels of cAMP produced by oocyte-specific adenylyl cyclase type 3 [81,82]. The protein kinase A (PKA) activated by cAMP plays a crucial role in maintaining the cell cycle arrest in oocytes [83,84,85,86,87]. However, it is not directly involved in regulating the key cell cycle engine, MPF, to arrest the oocyte cell cycle. In GV oocytes, PKA indirectly regulates MPF activity through Wee2 (Wee1B) kinase and Cdc25B. PKA promotes the activity of Wee2 kinase, which primarily phosphorylates and inhibits the Cdk1 component of MPF in the nucleus (Figure 3). When Wee2 is knocked down or its protein level is reduced by morpholino, GV oocytes are released from the cell cycle arrest [88,89,90,91]. Additionally, PKA inactivates cdc25 phosphatase, which promotes Cdk1 activity [92,93].

### 2.1. GV Arrest by Translational Regulation

MPF activity is regulated not only by protein modifications but also cyclin expression. Considering that transcription does not occur in fully grown oocytes [94], the translation of accumulated maternal mRNA is the sole gene expression mechanism underlying the stimulation of oocyte maturation in most species [95,96]. The translation of each mRNA within the oocyte is precisely and timely regulated for their role during the oocyte development [97].

Translation control in GV oocytes is little known in mouse oocytes. However, there are relatively many studies using *Xenopus* oocyte. The regulation occurs at the translational initiation stage. In *Xenopus* GV oocytes, elF4E binding to the 5’ cap structure of transcripts and various RNA binding proteins, (such as CPEB1, cleavage and polyadenylation specificity factor (CPSF), embryonic poly(A) binding protein (ePABP)), adenylase (Gld2), deadenylase (PARN), and scaffold proteins (symplekin, maskin) in the 3′ untranslated region (3′ UTR), form a complex that maintains a short poly(A) tail, thereby inhibiting translation [98,99]. This mechanism suppresses the translation of several key cell cycle regulators, including cyclin B1 and Mos, resulting in low MPF activity. Whether this complex functions similarly in translation regulation of mammalian oocytes including mouse oocytes is currently unclear.

Cyclin concentration in the GV state varies among species. For example, in *Xenopus* GV oocytes, one of the most extensively studied oocytes, the stockpile of cyclins is relatively low compared to the amount of Cdk1, resulting in insufficient generation of the Cdk1–cyclin B complex. Therefore, de novo protein synthesis is required for germinal vesicle breakdown (GVBD) to occur [100]. In contrast, mouse GV oocytes have a relatively high abundance of cyclins, and the Cdk1–cyclin B complex is preassembled [101,102]. In this case, the preassembled Cdk1–cyclin B complex is phosphorylated and inhibited by Wee2, preventing meiotic cell cycle re-entry of the oocyte (Figure 3). Recent studies have shown, using the RiboTag immunoprecipitation method that in mouse GV-arrested oocytes, that the *Ccnb2* transcript is more abundantly associated with ribosomes than the *Ccnb1* transcript. Consequently, cyclin B2, whose concentration exceeds that of cyclin B1, maintains the preassembled Cdk1–cyclin B complex [39]. Additionally, the presence of *Ccnb3* mRNA associated with ribosome complexes in GV oocytes suggests isotype translation [26].

Mouse GV oocytes express the cyclin A1 protein, but the mRNA levels are low. The mRNA of cyclin A2 is expressed in all tissues, including oocytes, but the protein is not detected in GV oocyte [35,37,61,71]. Cyclin O is highly expressed in oocytes [33]. However, research on the expression regulation of these cyclins in GV oocytes is very rare.

### 2.2. GV Arrest by Degradation

When the cyclin concentration in the GV oocyte is excessively high, the oocyte undergoes an unwanted cell cycle without a cell cycle resumption signal. Conversely, extremely low cyclin concentration complicates the rapid resumption of the cell cycle even when the related signal emerges. Therefore, GV stage oocytes maintain a delicate balance between cyclin synthesis and degradation to ensure an appropriate level of cyclins for prompt Cdk1 activation and cell cycle resumption [103]. In mouse prophase I arrested oocytes, cyclin B1 is predominantly present in the cytoplasm, with some cyclin B1 localized in the nucleus. The APC/C activated by Cdh1 promotes the degradation of cyclin B1, thereby maintaining low Cdk1 activity [104,105,106]. Cdc14B attenuates the inhibition of Cdh1 phosphorylation to regulate its activity, and BubR1 contributes to the GV arrest by maintaining high levels of Cdh1, leading to reduced Cdk1 activity [107,108].

Conversely, securin acts as a competitive substrate of cyclin B1 for APC/C^Cdh1^ degradation, and Emi suppresses APC/C^Cdh1^. Therefore, both help maintain an appropriate level of cyclin B1 [109,110]. Exogenous Emi1 or the inhibition of Emi destruction in GV oocytes leads to an accumulation of cyclin B1 that is sufficient to promote GVBD [111]. Similarly, the degradation of cyclin B2 is finely tuned by Hec1 (also known as Ndc80), a kinetochore protein and subunit of the Ndc80 complex, which acts as a co-competing substrate for APC/C^Cdh1^, thus regulating the amount of cyclin B2 in GV oocytes [112].

## 3. From Resumption of Oocyte Maturation to Metaphase I

The progression of the cell cycle from the prophase I arrested oocyte to metaphase II egg is as follows: (1) resumption of the cell cycle, accompanied by GVBD, chromosome condensation, and spindle formation (Figure 3 prometaphase I); (2) complete formation of the spindle and alignment of chromosomes in metaphase I (Figure 3 metaphase I); (3) entry into anaphase I and separation of homologous chromosomes, followed by extrusion of the first polar body (Figure 3 anaphase I); (4) the transition from meiosis I to meiosis II without an intervening S-phase; and (5) the metaphase II state, where the cell cycle is arrested again (Figure 3 metaphase II).

Gonadotrophic hormones stimulate the periodic ovulation of progressively growing oocytes within the ovary. Changes in the signaling pathways of various cells within the follicle, induced by luteinizing hormone, ultimately activate phosphodiesterase 3A (PDE3A) in the oocyte, leading to the degradation of cAMP [113,114,115,116,117,118]. Consequently, PKA is inactivated, allowing Wee2 to move out of the nucleus. Cdk1–cyclin B1 and Cdc25B translocate into the nucleus, leading to the breakdown of the GV membrane and resumption of the oocyte cell cycle [91,119,120]. The nuclear translocation of exogenous cyclin B1 promotes GVBD, suggesting the importance of the nuclear translocation of cyclin B1 for GVBD [121].

Cyclin B2 is located around the germinal vesicle in *Xenopus* oocytes, but is primarily present in the Golgi during mitosis [122]. In mouse oocytes, cyclin B2 translocates into the nucleus before GVBD, and cyclin B2 moves into the nucleus to compensate for the role of cyclin B1 in cyclin B1 knockout oocytes, indicating that it also regulates nuclear MPF activity [29]. Cyclin O is also present in the GV and localizes to the spindle region after GVBD [33]. However, the timing, sequence, and rate of translocation for each cyclin, and their specific substrates, have not been studied.

In addition to the regulation of MPF activity by Wee2-Cdc25B, kinases such as c-Mos and Polo-like kinase, as well as phosphatases, including PP1 and MASTL-Arpp19 (ENSA)-PP2A, regulate MPF activity through positive and negative feedback mechanisms. The regulation of MPF by these kinases and phosphatases has been extensively reviewed [123,124].

### 3.1. Role of Cyclins from Resumption of Oocyte Maturation to Metaphase I

The prometaphase I of oocyte meiosis is relatively prolonged compared to that of somatic cells. The slow progression during prometaphase allows for chromosome condensation, congression, kinetochore−microtubule attachment, and proper alignment of chromosomes at the metaphase plate [125]. This is crucial for accurate separation of homologous chromosomes and prevention of aneuploidy. The main driving force for GVBD and prometaphase progression is the increasing activity of Cdk1, which reaches its peak at metaphase I. Continuous synthesis and accumulation of cyclin B1 have been believed to be essential for increasing Cdk1 activity and meiosis progression [103,126]. However, recent studies have shown that *Ccnb1*^−/−^ oocytes undergo normal GVBD, complete meiosis I, and extrude the first polar body. Furthermore, no abnormalities in chromosome alignment and spindle formation are observed during meiosis I [27,29]. These findings suggest that cyclin B1 may not be required for the progression of meiosis I.

However, the upregulation of cyclin B2 by an unknown mechanism in *Ccnb1*^−/−^ oocytes results in complete MPF activation [29], which is also observed in mitosis [127]. Therefore, whether cyclin B1 is truly dispensable for GVBD and metaphase I processes or whether its significance is counteracted by compensation through cyclin B2 has not been determined. Recent studies suggest that cyclins forming complexes with Cdk1 are more stable than free cyclins [128]; therefore, there is a possibility that Cdk1 preferentially associates with cyclin B2 in the absence of cyclin B1, leading to increased stability and elevated levels of cyclin B2.

*Ccnb2* knockout mice are viable, but oocytes lacking cyclin B2 are unable to efficiently activate Cdk1, resulting in significant delays in GVBD or metaphase arrest owing to SAC activation. Consequently, these mice exhibit impaired oocyte maturation, early ovarian insufficiency, and compromised fertility (Figure 1) [27,29,39,112,129]. In vitro studies have shown that overexpression of cyclin B2 induces GVBD more efficiently than that of cyclin B1 in the presence of dbcAMP, an oocyte maturation inhibitor [103]. These findings demonstrate the necessity of cyclin B2 for sufficient MPF activation which is required for re−entry and progression of meiotic divisions. However, the fact that oocytes remain arrested in the GV state when both cyclin B1 and B2 are absent indicates that one of the two cyclins is essential for GVBD [29].

The onset of GVBD at normal timing in *Ccnb3*^−/−^ mice demonstrates that cyclinB3 is not required for the early stages of oocyte maturation [31]. Overexpression of cyclin A1 induces GVBD without cyclin B1 and B2 in the presence of the GVBD inhibitor milrinone [36,130]. Cyclin A2 is present in GV oocytes, and microinjection of stable cyclin A2 mutants lacking the destruction box induces spontaneous GVBD, while microinjection of cyclin A2 antibodies slows down the rate of GVBD [37]. Both endogenous cyclin A1 and A2 can bind to Cdk1, but their knockout does not affect GVBD, suggesting that they may not function during this stage in vivo [34,74].

Cyclin O in oocytes seemingly regulates GVBD through different mechanisms compared to other cyclins. Knockdown of *Ccno* using short interfering RNA microinjection prevents the dephosphorylation of the Cdk1 inhibitory phosphorylation site, Tyr15, leading to the oocytes remaining in the GV state. This GV arrest can be rescued by the overexpression of cyclin B1 but not Cdc25B, suggesting that it is not directly related to the Cdc25B activity. Whether the exact role of cyclin O in GVBD is inactivation of Wee2 has not been determined. In *Ccno*-knockdown oocytes, the formation of microtubule-organizing centers is disrupted in the perinuclear region [33]. This suggests that the primary role of *Ccno* in oocytes is not the regulation of MPF but rather the regulation of early spindle formation.

### 3.2. Translational Regulation of Cyclins

The continuous synthesis and accumulation of various proteins, including cyclin B1, are the main driving force for oocyte maturation. The mechanism regulating protein translation in mouse oocyte meiotic cells remains largely unexplored. However, the process has been relatively well-studied due to the requirement for mRNA translation during re-entry into meiosis in *Xenopus* oocytes [131,132]. Therefore, results gained from *Xenopus* studies could potentially offer clues regarding protein translation in mouse oocytes. The length of the poly(A) tail of mRNA is crucial for the regulation of translation, and relatively longer poly(A) tails (approximately 80–500 residues) are generally associated with increased translation. In *Xenopus* GV oocytes, the poly(A) tail of mRNAs encoding proteins, such as Mos and cyclin, is relatively short: approximately 20 nucleotides [133]. Upon stimulation by progesterone to initiate oocyte maturation, the poly(A) tails of mRNAs lengthen to approximately 100 nucleotides, leading to increased translation [131].

The regulation of mRNA poly(A) tail length involves two main cis-acting sequences: the hexanucleotide polyadenylation signal (AAUAAA) and the cytoplasmic polyadenylation element (CPE, UUUUUAU) [134]. The hexanucleotide polyadenylation signal is a highly conserved sequence located 10–30 nucleotides upstream of the mRNA’s cleavage/polyadenylation site [135]. It is identified by a complex of three or four proteins, including CPSF, which catalyzes the cleavage and polyadenylation of pre-mRNA [136]. The CPE site binds to the RNA-binding protein CPEB, a highly conserved protein that regulates the elongation of mRNA’s poly(A) tail. CPEB generally activates the translation of target mRNAs but can also act as a translation repressor depending on its phosphorylation state [137,138]. In *Xenopus* GV oocytes, CPEB binds to proteins, such as maskin, which inhibits the translation of CPE-containing mRNAs, and PARN, which possesses 3′-exoribonuclease activity, to maintain a short polyadenylated tail [139]. When translation is activated upon oocyte maturation, CPEB is phosphorylated by Eg2/Aurora A kinase [140], PARN is released from the inhibitory complex, and mRNA is polyadenylated by cytoplasmic poly(A) polymerase GLD-2 [141]. The PABP forms a complex with eIF4G of the eIF4F cap-binding structure at the 5’ end of mRNA, thereby promoting mRNA looping and enhancing translation [133,142,143,144,145]. CPE-containing mRNAs are not polyadenylated simultaneously with the initiation of oocyte maturation, but rather undergo translation at specific time points. For example, mRNA encoding proteins like Mos are polyadenylated early during prophase I, while mRNAs encoding cyclin B1 are polyadenylated late after GVBD. Early polyadenylation is induced by CPEB phosphorylation mediated by aurora A, while late polyadenylation requires Mos synthesis and cdk1 kinase to phosphorylate CPEB, with most CPEBs being degraded [146].

While the mechanisms regulating mRNA polyadenylation and translation during oocyte development in other vertebrates, including mice, have not been extensively studied, it is presumed that similar regulations occur. For example, in zebrafish and mice, polyadenylation of cyclin B1 mRNA occurs during oocyte maturation, and the hexanucleotide and CPE sequences are essential for mRNA translation [95]. However, the position and sequence of CPE in cyclin B1 mRNA 3′ UTR are not well conserved across species (Figure 4). This is likely because the factors and mechanisms that regulate the translation of maternal mRNAs are conserved, but the levels of cyclin proteins required for early oocyte maturation may differ among species.

The recent discovery of various isoforms of cyclin B in oocytes has increased interest in understanding their specific expression regulation and unique functions during oocyte maturation. In *Xenopus* oocytes, five types of cyclin B isotypes (B1–B5) have been identified, each with different expression patterns regulated during oocyte maturation. In GV oocytes, all five cyclin mRNAs have short poly(A) tails, and, upon progesterone stimulation, cytoplasmic polyadenylation occurs for cyclin B1, B2, B4, and B5, but not for B3 mRNA. Synthesis of cyclin B2 and B5 is required for the progression of early meiosis I, while new synthesis of B1 and B4 is needed for the progression of meiosis II [146,147].

There is a clear difference between the protein translation patterns of cyclin B1 mRNA and cyclin B2 mRNAs in mouse oocytes. In GV stage mouse oocytes, cyclin B1 expression is kept low through the regulation of CPEB and mRNA granule formation [148,149], while the expression level of cyclin B2 remains relatively constant during oocyte maturation [26]. Mouse oocytes can enter meiosis without de novo protein synthesis, likely due to the relatively high concentration of cyclin B2 [150]. The difference in the expression of cyclin B1 and B2 can be attributed to the distinct structures of their 3′ UTRs and the number and location of CPEs (Figure 4). When RNA immunoprecipitation using CPEB antibodies is performed to confirm CPEB binding to mRNAs, cyclin B1 transcripts are significantly coprecipitated, while cyclin B2 transcripts show minimal binding. Knockdown of CPEB1 using morpholinos had little effect on the expression of the cyclin B2 3′ UTR reporter, but significantly reduced the expression of the reporter containing cyclin B1 3′ UTR. Consequently, the translation of cyclin B1 mRNA is regulated by CPEB1, while cyclin B2 acts independently of CPEB1. Additionally, the regulation of cyclin B1 expression by CPEB is Cdk1-associated activity-dependent [26]. According to recent research, the presence or absence of cyclin B2 affects the translation of cyclin B1 mRNA [39]. This suggests that the Cdk1-associated activity involved in regulating CPEB could be the Cdk1–cyclin B2 complex, but further studies are needed to confirm this.

Cyclin B1 mRNA expression during oocyte maturation is also regulated by a newly identified mechanism. Cyclin B1 mRNA utilizes three alternative hexanucleotide poly(A) sites to generate three transcripts with varying lengths of the 3′ UTR in the oocyte (Figure 4). The transcript with a short 3′ UTR is expressed from the GV stage, while the intermediate and long 3′ UTR transcripts are suppressed in the GV stage, and their expression begins after GVBD (Figure 3) [151]. Recent studies have shown that cyclin B2 contains two alternative poly(A) sites, but their expression does not exhibit differences [39]. Whether the short form of the mouse cyclin B2 transcript is actually expressed in the oocyte has not been elucidated. Furthermore, the regulation of cyclin B1 and B2 expression is influenced by the 3′ UTR, 5′ UTR, and open reading frame [152]. Using these regulatory elements, the oocyte is able to control the precise timing and amount of cyclin B1 and B2 production.

The mRNA levels of cyclin B3 increase approximately 3–4 times during the metaphase compared to levels in the GV phase [30]. However, the amount of cyclin B3 mRNA associated with polysomes is very high in the GV state and gradually decreases with oocyte maturation [26]. Therefore, measurement of the precise amount and regulation of cyclin B3 expression during oocyte maturation is required for the examination of changes in protein levels.

### 3.3. Regulation of Cyclin Degradation

During the GV phase, the levels of cyclin B are regulated to a low state by APC/C^Cdh1^. However, during the GVBD stage, the degradation of cyclin B1 by APC/C^Cdh1^ is inhibited, leading to its increased accumulation. Recent research has revealed that the kinetochore complex component, Mis12, regulates cyclin B1 degradation by controlling Cdh1 or Cdc14B, thereby inducing the accumulation of cyclin B1 (Figure 3. prometaphase I). Depletion of Mis12 prevents the accumulation of cyclin B1, resulting in the inhibition of GVBD [153]. Additionally, CenpH, a component of the inner kinetochore protein, also modulates APC/C^Cdh1^ to enhance the accumulation of cyclin B1 at the GVBD stage. Therefore, depletion of CenpH by morpholino injection resulted in attenuation of MPF activation [110].

Once the spindle has been formed and microtubules attach to kinetochores, homologous chromosomes are aligned accurately at the metaphase plate. Only when the homologous chromosomes are properly positioned and aligned, can they be evenly divided into two cells. Until then, the cell cycle is arrested by the SAC triggered by improper attachment of kinetochores to microtubules. SAC inhibits APC/C^Cdc20^, resulting in the stabilization and accumulation of securin and the Cdk1–cyclin B1 complex. These components bind to and inhibit separase by phosphorylation [106,154,155,156,157,158,159].

Recent research has shown that cyclin B2 is also involved in these processes. In mouse oocytes, Cdk1–cyclin B2 binds to separase, and cyclin B2 degradation is necessary for separase activation during the metaphase I-anaphase I transition. Furthermore, in cyclin B1-null oocytes, the presence of stable cyclin B2 does not prevent chromosome separation if non-phosphorylatable separase is present, indicating that Cdk1–cyclin B2 phosphorylates separase to inhibit anaphase I progression (Figure 3. metaphase I) [129,160]. Oocyte-specific deletion of cyclin B2 leads to delayed spindle assembly and increased errors in chromosome segregation during the metaphase I- anaphase I transition, highlighting the important role of cyclin B2 in metaphase I of mouse oocytes [39].

## 4. Metaphase I-Anaphase I Transition

Metaphase I-anaphase I transition during oocyte maturation is primarily regulated by protein degradation. When the chromosomes are properly aligned and all microtubules are attached to the kinetochores, the SAC is inactivated, and APC/C^Cdc20^ promotes the polyubiquitination of securin and cyclin B1 through the ubiquitin-proteasome system for their degradation [158,161,162,163,164]. Overexpression or stable expression of cyclin B1 arrests oocytes at metaphase I [158], indicating that the degradation of cyclin B1 is essential for the metaphase I-anaphase I transition and the extrusion of the first polar body [162,165,166]. Recent studies have shown that cyclin B2 is also degraded after metaphase I; if cyclin B2 is not degraded, homologous chromosome separation does not occur, and the oocyte remains arrested at metaphase I.

Indeed, the degradation of cyclins relies on the presence of a destruction box, also known as the D box, and KEN box within the cyclin protein (Figure 2). B-type cyclins typically contain a conserved D box motif RXALGXIXN, where the arginine at position 1 and the leucine at position 4 are highly conserved. Mutations in the D box lead to the stabilization of cyclins and a decrease or lack of ubiquitination [167]. However, recent research has identified a novel degradation site that is involved in regulating the timing of spindle alignment. In the early prometaphase stage of oocytes, there are two groups of cyclin B1. One group is a pool of free cyclin B1, and the other is cyclin B1 bound to Cdk1. During progression to metaphase I, most of the free cyclin B1 is degraded through the newly identified degradation site (Figure 2A, Prometaphase motif, PM motif), while the Cdk1-bound cyclin B1 remains protected from degradation until SAC is satisfied and APC/C^Cdc20^ is fully activated. Only then does degradation of Cdk1-bound cyclin B1 begins, leading to a rapid decrease in Cdk1 activity and the initiation of anaphase I [128]. This mechanism allows oocytes extra time for the precise alignment of chromosomes. This motif was also found in the cyclin B2, indicating that cyclin B2 is possibly regulated by similar mechanism.

The role of cyclin B3 in the metaphase I-anaphase I transition has been elucidated by recent studies. Oocytes lacking cyclin B3 maintain the metaphase I spindle and arrest at metaphase I, resulting in female mouse infertility [31,32]. This suggests that cyclin B3 plays an important role in the metaphase I-anaphase I transition. Upon metaphase arrest by cyclin B3 removal, cyclin B1 remains stable, and Cdk1 activity is maintained in a high state, even when SAC is inactive. Therefore, cyclin B3 appears to regulate the degradation of cyclin B1 and securin by modulating APC/C activation [30]. The inability of cyclin B1 expression to compensate for the lack of cyclin B3 suggests that B3 plays a unique role at this stage.

The mechanism by which cyclin B3 promotes anaphase I onset was unclear until the recent revelation of the role played by Cdk1–cyclin B3 in metaphase. Cdk1–cyclin B3 phosphorylates the conserved site of Emi2 during meiosis I, leading to Emi2 phosphorylation by various kinases, including polo-like kinase 1, and degradation (Figure 3 metaphase I and anaphase I). Therefore, in the absence of cyclin B3 in mouse and *Xenopus* oocytes, the Emi2-dependent cell cycle arrest occurs prematurely at metaphase I, resulting in cell cycle arrest. Cyclin B3 is present at a sufficiently high level in early meiosis I and is degraded upon completion of meiosis I, preventing its re-accumulation [38]. Furthermore, in female fruit fly germ cells, the absence or mutation of cyclin B3 leads to an inability to complete meiosis I and infertility, suggesting the conservation of cyclin B3′s role across species [54,57,168].

The mechanisms underlying the sparing of cyclin B3 until completion of meiosis I and the selective degradation of only cyclin B1 and B2 during the metaphase I-anaphase I transition, and the mechanism underlying the distinction of the cyclins by APC/C^Cdc20^ can potentially be explained by the differences in the D boxes of each cyclin. Cyclin B1 and B2 have well-conserved D boxes, whereas cyclin B3 has a quasi-D box which contains phenylalanine instead of leucine at the conserved position 4 of the D box motif (Figure 2). This difference may lead to differences in affinity and specificity for APC/C, making cyclin B3 more resistant to degradation during the metaphase I-anaphase I transition than B1 and B2. Further experiments and research are needed to validate these possibilities and investigate the mechanisms underlying the degradation of cyclin B3 after anaphase I.

Cyclin A2 and cyclin O also possess putative D-box motifs (Figure 2). Cyclin A2 undergoes two waves of degradation during oocyte maturation, with most of cyclin A2 being degraded during prometaphase I. Overexpression of wild-type cyclin A2 inhibits PB extrusion in meiosis I, while constitutive cyclin A2 activity in meiosis I leads to precocious sister separation, suggesting that cyclin A2 degradation during prometaphase I is essential for oocyte meiotic progression [37]. The degradation of cyclin O in the mouse oocyte has not been reported.

## 5. Anaphase I-Metaphase II Transition

Anaphase I to metaphase II transition is relatively rapid, and therefore, research on the regulation of this stage is limited. Following the first polar body extrusion, Cdk1–cyclin B1 activity increases, preventing the onset of interphase or S-phase and facilitating entry into the second meiotic division (meiosis II). *Ccnb1*^−/−^ oocytes fail to progress to meiosis II after the first polar body extrusion and instead undergo chromosome decondensation and enter interphase. However, this defect can be rescued via the exogenous injection of mRNA encoding cyclin B2. This suggests that sufficient MPF activation, regardless of cyclin B1 or B2, is crucial for entry into meiosis II [27,29].

Given that oocyte-specific *Ccnb1*-null female mice are infertile due to MPF reactivation failure in meiosis II, the reason for the inability of endogenous cyclin B2 to compensate for the absence of cyclin B1 in *Ccnb1*^−/−^ mice is not fully understood. It may be because translation regulation mechanisms involving factors like CPEB may selectively enhance the translation of cyclin B1 in wild-type mice, while suppressing that of cyclin B2. Alternatively, the degradation of cyclin B2 and a decrease in the degradation of cyclin B1, or a combination of both factors may be the underlying reason. However, these possibilities and the regulation of cyclin B2 protein level in this step have not been extensively studied.

In *Drosophila* oocytes, cyclin B3 plays a role in inhibiting the entry into the S phase between the first and second meiotic divisions by cooperating with cyclin A [54]. In contrast, in *Xenopus*, the XeWee1A protein (ortholog of mouse Wee2) is absent, resulting in the maintenance of high Cdk1 activity and the inhibition of re-entry into the S phase between the first and second M phases [169,170]. In mice, Wee2 is present at this stage; however, the mechanism by which Wee2 activity is reduced is unknown [88,90,171].

For proper spindle formation during metaphase II, cyclin A2 is necessary [74]. Cyclin A2, which decreases during prometaphase I, is re-expressed and maintained until arrest of oocytes at metaphase II. Conditional deletion of cyclin A2 does not have a specific impact on the first meiotic division but leads to impaired metaphase II spindles and increased merotelic attachments. Therefore, cyclin A2 plays a role in spindle assembly during metaphase II and provides flexibility in kinetochore-microtubule attachments, allowing for error correction and prevention of merotelic attachments and lagging chromosomes [37]. However, the detailed mechanisms involved in this process have not been elucidated.

## 6. Arrest at Metaphase II and Fertilization

Prior to undergoing fertilization, oocytes remain arrested in the metaphase II phase owing to cytostatic factor (CSF) activity [1,172,173]. CSF activity is maintained by the mitogen-activated protein kinase pathway activated by Mos, which regulates MPF activity [174]. During the metaphase II phase, Cdk1 activity is maintained at a high level, and the balance between synthesis and degradation of cyclin B1 is crucial for maintaining a threshold (Figure 3 metaphase II) [175,176,177]. Synthesis of cyclin B1 has been observed during metaphase II arrest; at the same time, cyclin B1 is degraded by APC/C^Cdc20^ activity. If excessive degradation of cyclin B1 leads to a decrease in Cdk1 activity, another regulatory pathway called the Mos-MAPK pathway is activated, thereby inhibiting APC/C activity through components, such as Emi/Erp (Emi2, XErp1 in *Xenopus*), and resulting in the stabilization of cyclin B1 [36,178,179]. Both MPF and CSF activities are lost upon fertilization or parthenogenetic activation [180].

In *Xenopus laevis* oocytes, cyclin E1 and E2 form a cyclin E–Cdk2 complex, which acts as a CSF by inhibiting APC/C, thus contributing to metaphase II arrest [181]. Additionally, cyclin E is essential for the development of *Drosophila melanogaster* and *C. elegans*, but it is not required for normal mouse embryonic development [182,183,184]. In mammals, fertilization by sperm triggers a Ca^2+^ spike, leading to the activation of the oocyte. The main downstream target of Ca^2+^ oscillation is calmodulin-dependent protein kinase II (CaMKII, CaMK2A) [185,186,187]. CaMKII phosphorylates Wee2, leading to the inactivation of MPF and initiating resumption of meiosis and exit from metaphase II [90,188,189]. CaMKII also inactivates Emi2, which activates APC/C. Activated APC/C then targets the cohesin, which is holding sister chromatids together, and cyclin B1 for degradation [190,191], leading to sister chromatid segregation and anaphase II entry. The oocyte exits meiosis II, and two pronuclei, one from the oocyte and another from the sperm, are formed. These pronuclei fuse to form a diploid zygote with a complete set of chromosomes [192]. The expression of stable cyclin B2 prevents sister chromatid separation due to insufficient separase activity, leading to a failure in pronucleus formation. However, when a Cdk1-resistant phosphorylation site mutant separase (PM-separase) is injected, sister chromatid separation occurs. Therefore, the cyclin B/CDK1 complex contributes to inhibition of separase activity during the second meiotic division in oocytes [160].

In *C. elegans*, the absence of cyclin B3 hinders the segregation of sister chromatids during anaphase II. This phenotype is partially rescued when SAC is simultaneously knocked down, suggesting that the function of cyclin B3 is SAC-dependent [56]. Additionally, in *C. elegans*, cyclin B3 plays a crucial role in zygotic genome activation during the maternal-to-zygotic transition [193]. However, there is no research on the role of cyclin B3 in mouse metaphase II oocytes. Cyclin A2 also appears to be involved in sister chromatid separation in meiosis II, as interfering with its function inhibits this process [37]. Various cyclins are involved in meiosis II phase and the activation of oocytes by fertilization, but research on the roles and regulation of cyclins during these stages is relatively limited.

## 7. Discussion

Precise regulation of oocyte maturation is crucial because oocytes are relatively large and have a long cell cycle. The discovery of various types of cyclins expressed in oocytes, following the initial identification of cyclin B1 as a key regulator of oocyte maturation, indicates that the process of oocyte maturation is considerably more complex and finely regulated than previously understood. The reason why oocytes utilize such diverse cyclins is still veiled. Particularly, for cyclin B1 and B2, although they differ in the timing and quantity of synthesis, they can compensate for each other in most processes and have similar levels of degradation. However, the reasons for the presence of both cyclins in oocytes are unknown. Each cyclin likely phosphorylates specific substrates during oocyte maturation. Therefore, research aimed at identifying these cyclin-specific substrates is essential and may yield interesting results regarding the regulation of oocyte cell cycle progression. Furthermore, a clear understanding of how the expression and degradation of each cyclin is regulated in time and space is necessary. It is also important to elucidate the differences in Cdk1 activation induced by each cyclin.

Detailed studies on the relative localization of each cyclin in oocytes are required. Most studies on the localization of cyclins in oocytes have been based on the overexpression of cyclins fused with GFP, which does not accurately measure the temporal changes in the localization of endogenous cyclins. For example, it is necessary to determine whether there is a time difference in the nuclear translocation of cyclin B1 and B2 during the GVBD stage and, if so, what their respective roles are.

In oocytes, Cdk1 primarily functions as a cell cycle-regulating kinase, and all the cyclins, including cyclin B1, B2, B3, and A2, associate with this kinase and exhibit activity in oocytes. In this context, examining cyclin competition to bind to Cdk1 would be intriguing. Determining the cyclin with the highest affinity for Cdk1 at each oocyte cell cycle stage and answering other related questions would be a fascinating area of research. In addition, the reasons why cyclin B2 increases when cyclin B1 is deleted, and whether the absence of one cyclin results in the other cyclins compensating for its roles in all substrates, are unknown. Additionally, further research to identify the possible existence of currently unknown cyclins expressed in oocytes and their functions is needed.

## Figures and Tables

**Figure 1 ijms-24-13659-f001:**
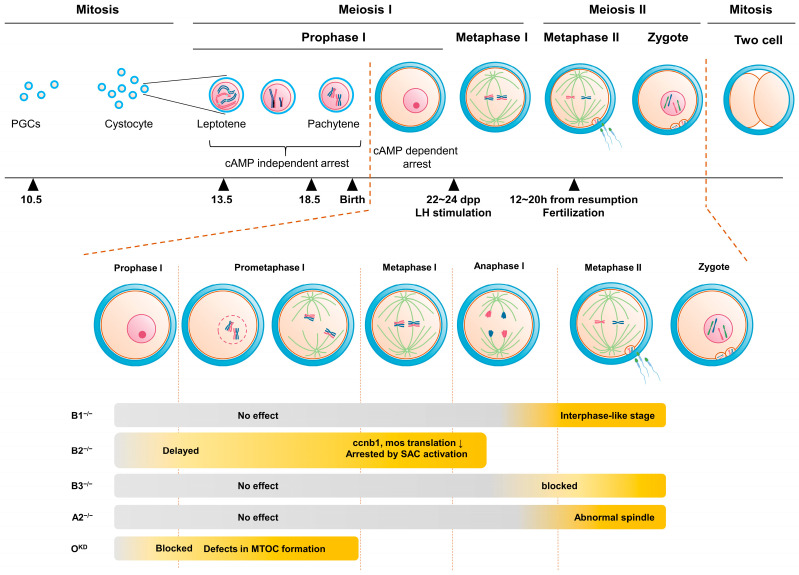
Oocyte maturation and impact of cyclin deletion. Mouse oocyte remains arrested in prophase I of the meiotic cell cycle until ovulation. At this stage, a distinctive large nucleus known as the germinal vesicle (GV) is observed in the oocyte, and upon resumption of the cell cycle triggered by luteinizing hormone the nuclear envelope disassembles in a process called germinal vesicle breakdown (GVBD). Subsequently, the oocyte proceeds through the first meiotic division, releasing the first polar body, and then progresses to metaphase II of the second meiotic division, and the oocyte is arrested again until fertilization. Upon fertilization, the second meiotic division is completed, leading to the release of the second polar body, followed by zygote formation and mitotic cell cycle. This process is primarily governed by the Cdk1–cyclin complex, commonly known as MPF. Recently, various cyclins have been identified in mouse oocytes. Deletion of each cyclin at specific stages of oocyte maturation results in defects, underscoring their unique roles in the process.

**Figure 2 ijms-24-13659-f002:**
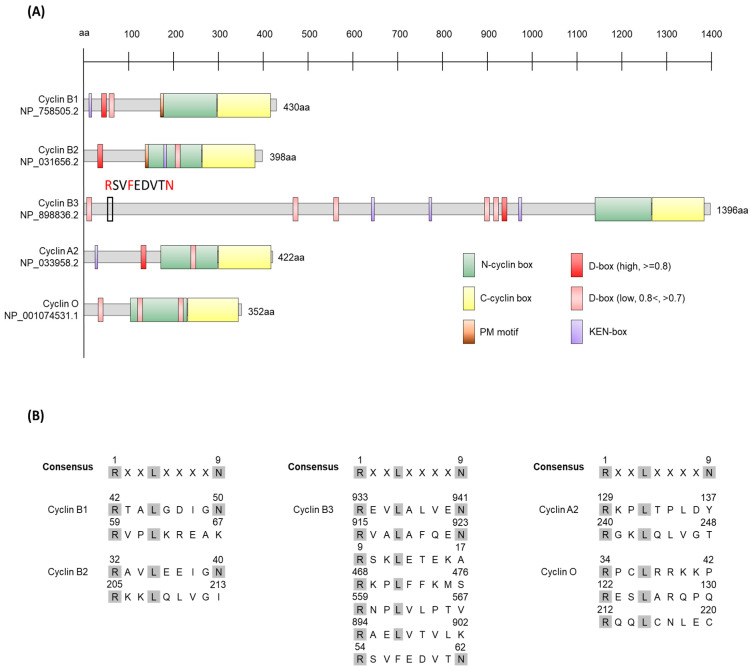
Structural features of cyclins found in oocytes. (**A**) Mouse cyclin proteins commonly possess a cyclin-box domain, a characteristic structural motif. Additionally, sequences containing the D box and KEN box motifs, which facilitate protein degradation, are also present. These motifs are crucial for regulating protein levels during the cell cycle. The D box and KEN box motifs were predicted at http://slim.icr.ac.uk/apc/index.php (accessed on 10 August 2023), with a similarity score threshold of 0.8. Recently, a new degradation site (PM motif) has been identified in the cyclin B1 and B2. This motif is involved in regulating the timing of spindle alignment by allowing free cyclin B to be degraded earlier during progression to metaphase I. The cyclin accession numbers utilized for the analysis are indicated on the left. (**B**) The comparison of D-boxes found in various cyclins. The D box has a highly conserved (especially arginine and leucine at 1 and 4 position) RXALGXIXN motif. Cyclin B3 has a quasi-D box, which contains phenylalanine instead of leucine at the conserved position 4 of the D box.

**Figure 3 ijms-24-13659-f003:**
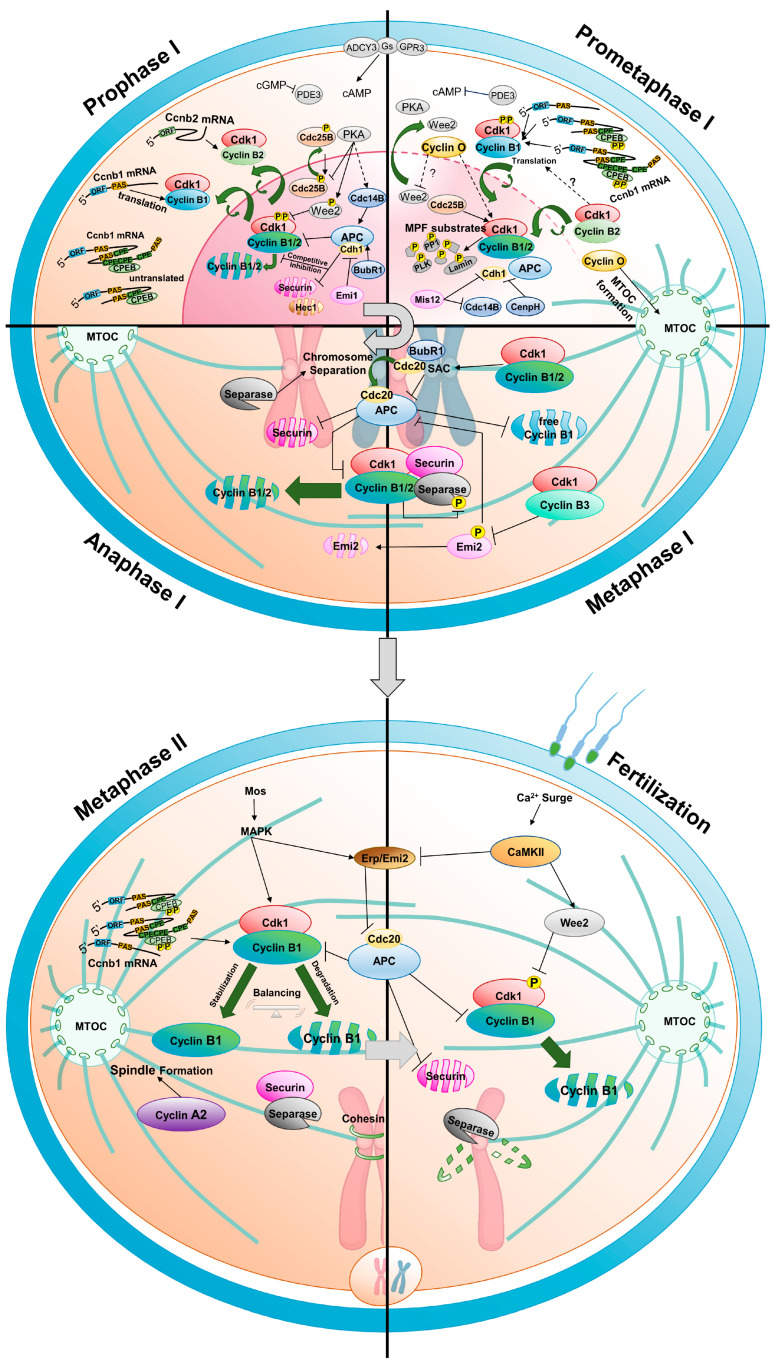
Roles of cyclins in the oocyte maturation process. (Prophase I) Maturation-promoting factor (MPF), composed of Cdk1 kinase and cyclin B, governs cell cycle arrest and resumption. Maternal mRNA is translated selectively. Cyclin B2 expression persists during prophase I arrest, while only the shortest cyclin B1 transcript translates, regulated by cytoplasmic polyadenylation element (CPE) in the 3′ UTR via CPEB. MPF remains inactive due to active Wee2, which is phosphorylated by cAMP-PKA. BubRI, Emi1, Hec1, and securin control APC^Cdh1^ activity and APCcdh1-mediated cyclin degradation maintains levels, ensuring controlled progression without undesired resumption or arrest (Prometaphase I). Cell cycle resumes by inactivating PKA and Wee2. Activated Cdc25B triggers GVBD with mainly cyclin B2-based MPF. Cyclin B1 transcripts, the intermediate and longest, translate, elevating cyclin B1 protein levels. APCCdh1 activity is modulated by Mis12 and CenpH. Cyclin O’s function in Cdk1 inhibitory phosphate removal and MTOC formation is unclear. (Metaphase I and Anaphase I) MPF and SAC inhibit APC^cdc20^ and separase until proper spindle alignment. Residual APC^cdc20^ degrades free cyclin B1, extending metaphase I. SAC inhibition activates APC^cdc20^, degrading cyclin B1 and B2, and lowering MPF activity. Cyclin B3 phosphorylates Emi2, promoting anaphase I (Metaphase II and Fertilization). After meiosis I, S phase-free cell division ensues. Anaphase I-metaphase II transition increases Cdk1-cyclin B1 activity. Cyclin B1 absence leads to interphase re-entry, unique to this stage. Metaphase II oocytes arrest via cytostatic factor (CSF), regulated by MPF and mitogen-activated protein kinase pathway. Fertilization activates APC/C and Wee2 via CaMKII, reducing MPF and enabling zygote’s mitotic cell cycle entry.

**Figure 4 ijms-24-13659-f004:**
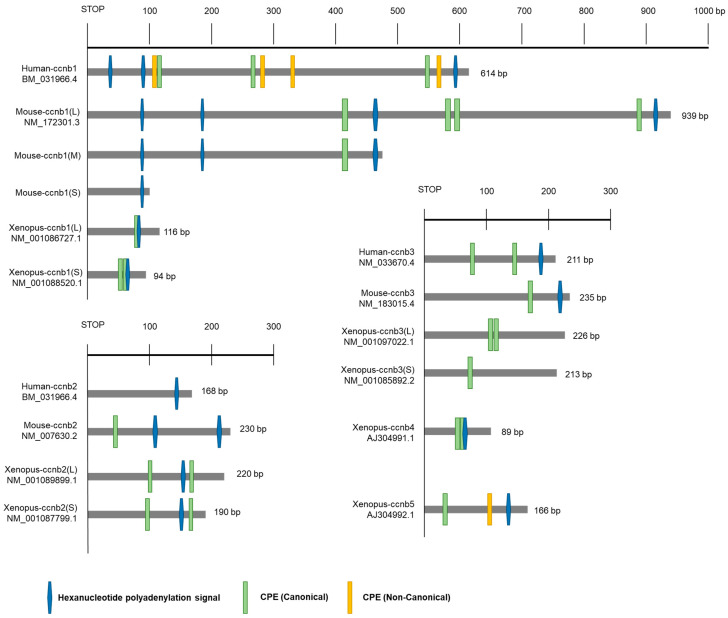
Structural motifs in the 3′ untranslated regions (UTRs) of cyclin B transcripts in mouse, human, and *Xenopus*. The 3′ UTR architecture of Ccnb gene orthologs varies across species, including mice, humans, and *Xenopus*. Horizontal bars represent sections of the 3′ UTR from each stop codon, with lengths indicated in base pairs (bp). Distinct hexanucleotide motifs within the same transcript generate mRNAs with different 3′ UTR lengths. Currently known examples include mouse *ccnb1* with three transcripts. Transcripts exhibit varying lengths of polyadenylation at different time points based on the structure of the 3′ UTR, thereby regulating translation. The structural motifs include hexanucleotide polyadenylation signal, canonical cytoplasmic element (CPE) which has well-conserved sequence, and non-canonical cytoplasmic element (CPE-NC) which has some modification.

## Data Availability

No new data were created in this study. All the data reported in this review were found in original articles cited in the text.

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
