# Peer review of "Meiotic Cell Cycle Progression in Mouse Oocytes: Role of Cyclins"

_ijms, 2023, doi:10.3390/ijms241713659_

Round 1
Reviewer 1 Report
This is a well written and comprehensive review that highlights in detail the expression and role of various cyclin proteins during oocyte maturation, of varied species. I only have a few comments/suggestions. 1) It would be helpful, though not essential, to show a graph of when the various cyclin isoforms are expressed at high levels and are degraded relative to the major events of oocyte maturation (i.e. GVBD, MII). 2) Page 6, second paragraph: all of this only applies after the oocyte becomes meiotically competent. Before that, maintenance of meiotic arrest is cAMP-independent. This should be specified. 3) The legend for Figure 3 is quite long and could be shortened because most of the information in the legend is also in the text. This figure is also complicated and could be simplified.
Reviewer 2 Report
The review work by Kim et al., described the role of different cyclins in regulating meiotic cell cycle in mice oocytes. The authors covered a large ground of literature and the manuscript is timely and well written.
I have the following few comments for the authors to consider.
1. The major comment for this review is that in many cases the authors used Xenopus work to establish certain concept. For example, in Section 2.1. GV arrest by translational regulation. Or in 3.2. Translational regulation of cyclins. While I understand that the majority of the oocyte maturation work is discovered in Xenopus and other model system, the authors should focus on mice oocyte according to the title of the manuscript. This will keep the theme of the manuscript intact.
In this regard, I have two suggestions: 1- The authors might mention that the data for that concept/process are primarily obtained from Xenopus and the work in mice is following up, or confirmed Xenopus work or failed to confirm, or not studied at all. Hence the authors are describing in that way in this section. And 2- The authors might cite works from Xenopus or other model system only after describing the work in mice (as written in Section 3. From resumption of oocyte maturation to metaphase I).
2. L 67-68- In many organisms, oocytes are mostly in a state of cell cycle arrest and can only exit this arrest state by specific hormonal signals [15]. The reference cited here is from human and the work is from 6 years back 2017. I suggest adding recent review that covers multiple organisms (e.g., Das and Arur, 2022. Regulation of oocyte maturation: Role of conserved ERK signaling)
3. L 90-92- However, recent studies have reported the expression of various cyclins, including cyclins B2, B3, A1, A2, E, and O, in oocytes of various species, such as Xenopus, Danio, mice, and humans. Need primary references for all organisms!
4. L 94- previously thought to have minor functions compared to cyclin B1. Need reference either the primary work or a review stating this view.
5. L 185-187- The author should add a developmental time-point of the oocyte Figure 1. E.g., when the oocytes are born in utero in mice, arrest period and then resuming meiosis. This will justify the use of Fig. 1 here and also provide a clear concept of this paragraph to the reader.
6. L 200-201- Since the authors focused primarily on mice oocyte, reference 87 from Xenopus oocyte is misleading. The author might use review paper here.
7. L 266- I strongly suggest using the term oocyte rather than egg.
Round 2
Reviewer 2 Report
The revised version satisfied all my queries.
Thanks